# *Entamoeba histolytica*: Membrane and Non-Membrane Protein Structure, Function, Immune Response Interaction, and Vaccine Development

**DOI:** 10.3390/membranes12111079

**Published:** 2022-10-31

**Authors:** Nurhana Jasni, Syazwan Saidin, Wong Weng Kin, Norsyahida Arifin, Nurulhasanah Othman

**Affiliations:** 1Institute for Research in Molecular Medicine (INFORMM), Universiti Sains Malaysia, Gelugor 11800, Malaysia; 2Department of Biology, Faculty of Science and Mathematics, Universiti Pendidikan Sultan Idris, Tanjung Malim 35900, Malaysia; 3School of Health Sciences, Universiti Sains Malaysia, Kubang Kerian 16150, Malaysia

**Keywords:** biological membrane, structure, function, immune response, *Entamoeba histolytica*, vaccine

## Abstract

*Entamoeba histolytica* is a protozoan parasite that is the causative agent of amoebiasis. This parasite has caused widespread infection in India, Africa, Mexico, and Central and South America, and results in 100,000 deaths yearly. An immune response is a body's mechanism for eradicating and fighting against substances it sees as harmful or foreign. *E. histolytica* biological membranes are considered foreign and immunogenic to the human body, thereby initiating the body's immune responses. Understanding immune response and antigen interaction are essential for vaccine development. Thus, this review aims to identify and understand the protein structure, function, and interaction of the biological membrane with the immune response, which could contribute to vaccine development. Furthermore, the current trend of vaccine development studies to combat amoebiasis is also reviewed.

## 1. Introduction

Amoebiasis is a parasitic infection caused by *Entamoeba histolytica* and constitutes an alarming public health concern in developing countries [1,2]. This parasite has recorded the highest rates of infection in Central and South America, Africa, and India [2,3]. Children, immigrants, travelers returning from endemic countries, patients with HIV, and homosexual men are among the groups with the most significant risk of infection [4,5,6]. The most common infection route is from ingesting contaminated food and water. Around 90% of the infected individuals are asymptomatic, whereas the other 10% show a wide range of clinical manifestations, such as colitis, dysentery, and extraintestinal amoebiasis [7,8,9]. The factors that turn the parasite virulent in asymptomatic individuals are still uncertain; however, the gut microbiota is considered one of the factors triggering the virulence [10]. Metronidazole has been the drug of choice against amoebiasis for decades; however, the toxic effects of metronidazole and recent failures in treating several intestinal protozoan parasites have led to a search for other drugs and mechanisms to combat this parasite [11,12].

The innate and adaptive immune systems work together to eliminate this parasite naturally. The innate immune system is the body's first defense mechanism against germs and is non-specific. The acid in the stomach and a thick layer of mucin mucus are examples of innate immune responses that amoebas will first encounter during the invasion [13,14]. Intrusive parasite control includes the secretion of cytokines or chemokines, the recruitment of immune cells, such as neutrophils and macrophages, and inflammasome activation [15]. An adaptive immune response is highly specific to the antigen that induced them, and the response is ‘remembered’ by the immune system, resulting in long-lasting protection [16,17]. Adaptive immune responses toward *E. histolytica* invasion include the generation of specific IgG antibodies [13,14,18]. *E. histolytica* biological membrane and non-membrane proteins are the molecules responsible for triggering immune responses. These membrane proteins are galactose/N-acetylgalactosamine (Gal/GalNAc) lectin, cysteine proteinase, thioredoxin, lipopeptidophosphoglycan, and protein disulfide isomerase. Meanwhile, the non-membrane proteins include *E. histolytica* ubiquitin, calreticulin, *E. histolytica* migration inhibitory factor (EhMIF), enolase, actins, alcohol dehydrogenase, and extracellular vesicle proteins.

As more is learned about the pathogenesis of *E. histolytica* infection and the host's immune reaction, the potential for developing an effective vaccine holds promise [3,18]. Modern subunit vaccines rely on combining a highly purified antigen with an adjuvant to increase their efficacy [4]. A successful vaccine must have the ability to induce long-term protective immunity and should involve a simultaneous response of antibodies and T cells [19,20]. Vaccination using native and recombinant Gal/GalNAc lectin was found to be successful in protecting animals against intestinal and amoebic liver abscesses [18]. However, no clinical trial validates its human efficiency [18]. To the best of our knowledge, there have been no licensed vaccines for the prevention of amoebiasis until now. Currently, work is in progress to develop a vaccine, and recent experimental studies seem promising.

Thus, this review outlines the protein structure, function, and interaction of immune response with the *E. histolytica* biological membrane and vaccine studies. The study was conducted by reviewing research publications from Google Scholar, PubMed, and Scopus. The keywords used were *E. histolytica* protein and antibody, structure, function, antigenic proteins, immune responses, and vaccine studies.

## 2. Biological Membrane Proteins and Antigens

Membrane proteins contain a signal peptide and transmembrane domain [21]. The proteins in the membrane can be divided into integral, peripheral, and lipid-anchored proteins [22,23]. The integral membrane protein is permanently anchored in the phospholipid bilayer. Meanwhile, peripheral membrane proteins are temporarily attached to a lipid bilayer or other integral proteins. Some *E. histolytica* biological membrane proteins are antigenic and were reported to induce a host immune response. These membrane proteins and components are Gal/GalNAc lectin, cysteine proteinase, EhLPPG, thioredoxin, and protein disulfide isomerase.

### 2.1. Gal/GalNAc Lectin 

Researchers widely study galactose/N-acetylgalactosamine (Gal/GalNAc) lectin as a membrane protein. It plays several roles in the pathogenicity of intestinal amoebiasis, including adherence to the host cell, cytotoxicity, complement resistance, induction of encystation, and generation of the cyst wall [24,25]. This protein contains a signal peptide and transmembrane domain; thus, it can be categorized as a membrane protein. The molecular weight of this protein is 260 kDa, and all *E. histolytica* strains express this protein on their surface [26]. Structurally, this protein is a heterotrimer composed of heavy (Hgl), intermediate (Igl), and light (Lgl) subunits [27,28,29]. The Hgl and Lgl are covalently attached to each other and interact non-covalently with the Igl. The Igl, which is the intermediate subunit, is constitutively localized at the lipid rafts; meanwhile, the heavy subunits (Hgl) and light subunits (Lgl) are transiently associated with rafts [30]. The Hgl has lectin activity; meanwhile, Lgl does not [28]. The intermediate subunits Igl have been shown to have hemolytic and cytotoxic activities that reside in the C-terminus of the protein [31].

The Gal/GalNAc lectin is antigenically conserved and can be characterized as immunogenic [26]. It is also considered to be the starting point of the invasion of *E. histolytica*, via attachment to the host mucin and colonic epithelial cells. The mucin forms high-affinity binding with the Gal/GalNAc lectin, allows the parasite to colonize the gut, and concurrently acts as a physical barrier [13,15].

In macrophages, *E. histolytica* Gal/GalNAc lectin up-regulates the expression of the mRNA of different cytokines [15]. Furthermore, it induces the mRNA expression of pattern recognition receptors, such as TLR-2 and TLR-4, modulated by nuclear factor NF-κB and the MAPK pathway [15]. The binding of Gal/GalNAc lectin to the TLR2 leads to NF-κB activation and the release of pro-inflammatory cytokines [32]. The lectin activates the CD4 cells, natural killer T cells, and CD8 T cells, enhancing protective cellular immunity [32].

The CD4 T cells secrete IL-13, IFN-γ, IL-4, and IL-5. Meanwhile, the CD8 T cells secrete the IL-17 cytokines. This IL-17 will induce the infiltration of neutrophils and enhance the secretion of antimicrobial peptides, mucin, and IgA into the colonic lumen [32].

Gal/GalNAc lectin and its components are widely used in vaccine development. Vaccines are biological products that elicit an immune response to a specific antigen, derived from an infectious disease-causing pathogen. Abhyankar et al. (2017) [4] evaluated the nanoliposome adjuvant system in triggering a strong mucosal immune response to the Gal/GalNAc lectin Lec A antigen of *E. histolytica* by immunizing mice with synthetic TLR agonists—alum, emulsion, or liposome formulations. Their results showed that the formulation of liposomes containing TLR4 and TLR7/8 agonists could generate intestinal IgA, plasma IgG2a/IgG1, IFN-γ, and IL-17A. This finding suggested that the nanoliposome delivery system, combined with TLR agonists, was promising for vaccine development against enteric pathogens.

In a subsequent study, Abhyankar et al. (2018) [33] formulated a stable PEGylated liposomal adjuvant containing synthetic GLA (TLR4) and 3M-052 (TLR7/8) ligands, which they administered intranasally to mice along with the Lec A antigen. According to their findings, the liposome containing both GLA and 3M-052 could elicit the secretion of Lec A-specific fecal IgA and Th1 immune responses. In addition, they conducted studies that modified the liposomal formulation at the PEG length. The optimized liposome increased the murine models' fecal IgA, serum IgG2a, systemic IFN-γ, and IL-17A levels. Furthermore, the adjuvant's improved formulation could reduce parasite antigens in the colon by more than 80%, which helped to protect against disease. The dose and excipient composition of each vaccine formulation component was optimized using the statistical design of experiment (DOE) and desirability approach, according to recent studies by Abhyankar et al. (2021) [1]. The optimization was conducted to maximize the desired characteristics while maintaining the physicochemical stability of the vaccine. This method has allowed for identifying the GLA and 3M-052 compositions, which exhibit high durability and protective immunity in the mouse model.

In a study conducted by Martinez-Hernandez et al. (2017; 2021) [34,35], the newly developed recombinant chimeric vaccine (PEΔIII-LC3-KDEL3) produced from the LC3 fragment of *E. histolytica*, domains I and II of *Pseudomonas aeruginosa*, and the carboxy-terminal signal KDEL3 in *Pichia pastoris* has potential as an immunogen and showed the inhibition of the adherence of trophozoites to the HepG2 cell monolayer in a hamster model. Furthermore, the vaccine protects against liver tissue damage and uncontrolled inflammation. Meneses-Ruiz et al. (2011; 2015) [36,37] demonstrated oral and intramuscular immunization using the *Autographa californica* baculovirus; expressing the *E. histolytica* Gal/GalNAc lectin LC3 fragment (AcNPV-LC3) could help to protect against ALA development in hamsters.

Next, the 150-kDa intermediate subunit (Igl) of the Gal/GalNAc lectin is gaining more attention in numerous studies and has been shown to have hemolytic activity [38]. Min et al. (2016) [39] studied the effect of immunization with recombinant Igl on amoebic liver abscess (ALA) formation. Their findings showed that recombinant Igl, particularly the C-terminal fragment, represents a promising vaccine against amoebiasis. It produced a significant humoral immune response with high antibodies, inhibiting 80% of the trophozoites’ adherence to mammalian cells and inducing the complement-mediated lysis of 80% of the trophozoites. Thus, the result shows that this subunit should be considered in vaccine development. The predicted protein structure of Gal/GalNAc lectin Igl1 can be found in Figure 1.

### 2.2. Cysteine Proteinase

Cysteine proteinases are virulence factors in *E. histolytica*; some of them were predicted to have transmembrane and glycosylphosphatidylinositol anchor attachment domains. Thus, they can be classified as membrane proteins [42,43]. This cysteine proteinase has several types; for example, EhCP1 and EhCP4 were found to colonize the large cytoplasmic vesicles that differ from the sites of other proteinases [43,44]. Fifty genes encode cysteine proteinases; *ehcp-a4* is the gene that is most up-regulated during the invasion and colonization of *E. histolytica* in the mouse cecal model [43]. Furthermore, the protein plays a role in tissue invasion by disrupting the colonic epithelial barrier, disrupting host defenses, and digesting extracellular matrix components such as immunoglobulins, complement, and cytokines [43,44,45]. Furthermore, cysteine proteinases also form a complex heterodimer protein (EhCPADH) by a combination of *E. histolytica* cysteine protease (EhCP112) and adhesin (EhADH), which is involved in the cytopathic mechanism and virulence of the parasite [46,47]. The EhCP112 alone can be inhibited by thiol inhibitors, such as E-64, TLCK, and chymostatin [47].

Meanwhile, a recombinant EhCP4 can be inhibited by a vinyl sulfone inhibitor, WRR605 [44]. The inhibition of EhCP4 reduced the parasite burden and inflammation in the mouse cecal model [44]. Cysteine proteinase 5 (EhCP5) was reported as a virulence factor in live *E. histolytica* that elicits a fast release of mucin [48].

The pro-mature cysteine proteinase 5 (PCP5) that was usually secreted or found on the surface of the amoeba stimulated NF-κB-mediated pro-inflammatory responses by the binding of the RGD motif to the αVβ3 integrin on Caco-2 colonic cells [49]. The predicted protein structure of cysteine proteinase 5 and its model confidence can be found in Figure 2.

### 2.3. Entamoeba Histolytica Lipopeptidophosphoglycan (EhLPPG) 

*E. histolytica* lipopeptidophosphoglycan (EhLPPG) is one of the virulence factors of *E. histolytica*; this macromolecule can promote tissue invasion by causing inflammatory damage to the host cells. It can be found on the *E. histolytica* membrane [51]. This molecule consists of 85% carbohydrate, 8% peptide, 2.5% lipid, and 1% phosphate [52] Moody et al. (1997) [53] and Wong-Baeza (2010) [52] reported the under-expression of LPPG and LPG in the non-virulent strain of *E. histolytica*, compared to the virulent strain. No LPG, and a modified version of LPPG, were found in the low-virulence Rahman strain and the non-pathogenic *Entamoeba dispar*, respectively. Thus, it can be inferred that the protein expression of LPPG and LPG influenced the virulence of the *Entamoeba* species.

In the *E. histolytica* immune responses, the interaction of EhLPPG with TLR2 and TLR4 resulted in the activation of NF-κB and the release of IL-10, IL-12p40, TNF-α, and IL-8 from human monocytes [54]. Furthermore, the purified phosphoinositol moiety of EhLPPG is capable of inducing IFN-γ production [51]. In particular, a diacylated PI, (1-O-[(28:0)-lyso-glycero-3-phosphatidyl-]2-O-(C16:0)-Ins) is a primary component of EhLPPG that is responsible for the activation of natural killer T cells [51].

In addition, the activation of dendritic cells by LPPG increased the expression of co-stimulatory molecules CD80, CD86, and CD40. Besides this, it produces pro-inflammatory cytokines, such as TNF-α, IL-8, and IL-12 [55]. This molecule can also induce antibody production and the formation of in vitro neutrophil extracellular traps (NET) [52,56].

### 2.4. Protein Disulfide Isomerase

Protein disulfide isomerase (PDI) is one of the antigenic membrane proteins found in *E. histolytica* [57]. The predicted protein disulfide isomerase structure and its model confidence can be found in Figure 3**.** This protein is classified as a membrane protein as it contains a transmembrane and signal peptide, as indicated by the Uniprot database and in studies reported by Azmi and Othman in 2020 [21]. It was also up-regulated in the membrane fractions [21]. Structurally, PDIs share a common feature associated with at least one active thioredoxin-like domain [58]. The predicted size of the protein is 41.73 kDa, with a length of 368 aa. The functions of this protein are to catalyze the oxidation, reduction, and isomerization of disulfide bonds in polypeptide substrates [59,60]. Furthermore, it also exhibits chaperone-like activity and can be inhibited by bacitracin [60]. The recombinant type of EhPDI shows the specific properties of PDI enzymes, such as performing oxidase and reductase activities [59]. This protein also participates in the cellular mechanism related to protein homeostasis [61]. For example, during stressful conditions, this enzyme is involved in holding, refolding, and degrading unfolded or misfolded proteins [58]. It also helps to protect heat-labile enzymes against thermal inactivation [58].

Mares et al. (2015) [58] used the PDI model of 38 kDa polypeptide with two active thioredoxin-like domains to study protein folding and disulfide bond formation in *E. histolytica*. Their studies revealed a significant difference in the functional role of each thioredoxin-like domain. Thus, their study indirectly indicated that the amount or type of thioredoxin domain might influence the PDI functions. 

Next, this protein was classified as antigenic by Kumarasamy et al. (2020), as this protein was recognized by sera from patients with an amoebic liver abscess [57]. However, no other study of immune response interaction with this *E. histolytica* protein could be found. Only a study of protein disulfide isomerases from other organisms was reported. One example was the immunization of BALB/c mice, with *Leishmania donovani* protein disulfide isomerase (LdPDI) as a DNA vaccine that elicited protective immunity against visceral leishmaniasis through the production of two pro-inflammatory cytokines, CD8 and CD4 (Th1 and Th17) [62]. Despite the lack of study on this protein, it merits further investigation as a drug target for anti-amoebic therapy and as a vaccine candidate.

### 2.5. Thioredoxin 

Thioredoxin is an antigenic protein that has been found in the *E. histolytica* membrane (Figure 4) [57]. Structurally, it contains a Toll/IL-1R/Resistance (TIR)-like domain and is predicted to have a signal peptide [64]. This protein is associated with thioredoxin reductase, forming the thioredoxin system. The two work together and play essential roles in *E. histolytica*. Thioredoxin is also one of the proteins that form an adduct with metronidazole metabolites and other proteins: thioredoxin reductase, superoxide dismutase, purine nucleoside phosphorylase, and a previously unknown protein [65]. Thioredoxin is crucial as it has various functions: acting as an antioxidative defense, modulating the intracellular and extracellular signaling pathways, cell growth, regulating DNA synthesis and transcription factor, modulating the immune response, apoptosis, and involvement in post-translational redox modifications on the target protein [57,66,67]. The silence of thioredoxin transcripts decreased the phagocytosis of erythrocytes and *Escherichia coli* [64]. The roles it fulfills make it suitable for use as a drug target or vaccine candidate, although no human immune response study was reported. Further study is suggested to elucidate its immune response interactions and potential as a vaccine candidate. 

## 3. Non-Membrane Protein

Non-membrane proteins have been found in other parts of *E. histolytica*, such as the cytoplasm and nucleus. It does not contain a signal peptide or transmembrane domain and thus excluded as a membrane protein. However, recent findings indicated that the *E. histolytica* cytosolic proteins were present in the membrane fraction and a few of them were antigenic [21,57]. Non-membrane proteins also contribute to initiating host immune responses. These proteins are ubiquitin, calreticulin, *E. histolytica* migration inhibitory factor (EhMIF), actin, alcohol dehydrogenase, enolase, and extracellular vesicles.

### 3.1. Entamoeba Histolytica Ubiquitin (Ehub)

Ubiquitin is one of the proteins found in *E. histolytica* and can trigger immune responses to produce antibodies. It is also immunogenic and antigenic [69]. According to the UniProt database, this protein can be found in the cytoplasm and nucleus of the parasite. The ubiquitin structure consists of protein moieties and glycosylated structures [69]. The sequence of *E. histolytica* ubiquitin amino acids, as deduced from a cDNA nucleotide sequence, was found to deviate by six positions from the consensus of all other known ubiquitins, the sources of which range from *Trypanosoma cruzi* to *Homo sapiens* [70]. Furthermore, the ubiquitin is also usually associated with the proteasome forming the ubiquitin-proteasome system, which is important for numerous cellular processes and for maintaining parasite homeostasis [69,71]. This ubiquitin-proteasome pathway presents a viable therapeutic target [72].

Infection with *E. histolytica* in humans induces strong IgG antibodies toward ubiquitin [69]. Human antibodies can recognize ubiquitin’s protein moieties and glycosylated structure [70]. Notably, the ubiquitin’s glycan moieties are immunodominant and induce IgG. Furthermore, antibodies to the ubiquitin Ehub are induced exclusively in patients with invasive amoebiasis and mainly to glycoprotein [73]. It indicates that glycan is immunodominant and could be a potential target for diagnostic tests, drugs, and vaccines to combat parasitic diseases [69,73].

### 3.2. Calreticulin

Calreticulin (CRT) and its model confidence, as shown in Figure 5, is one of the immunogenic proteins in *E. histolytica* that is localized in the endoplasmic reticulum [74,75]. It can also be found in the membrane fraction [74,75]. It is essential for regulating critical cellular functions, such as migration, apoptotic cell phagocytosis, and cytotoxic T lymphocyte- or natural killer T cell-mediated lysis [76]. It is also crucial in properly folding and transporting the protein through the endoplasmic reticulum [76]. CRT also interacts with human C1q and inhibits the activation of the classical complement pathway [77]. Both can be observed in the CRT of *E. histolytica* and *E. dispar* [77]. Furthermore, a higher level of CRT expression in the in-situ lesions of amoebic liver abscess (ALA) in the hamster model was found in *E. histolytica* compared to *E. dispar*. Additionally, CRT plays an important role in the early stages of the host-parasite relationship, in which the parasite needs to adapt to a new environment [74].

EhCRT is also an excellent immunogen for activating specific peripheral blood mononuclear cells (PBMC) [75]. PBMCs are a variety of immune cells, such as lymphocytes, monocytes, and dendritic cells, that works together to protect humans from harmful pathogens. EhCRT helps determine the immune response by inducing the differential expression of Ils, depending on the outcome of the disease. For example, patients with an amoebic liver abscess (ALA) during its acute phase (AP-ALA) show a Th2 cytokine profile; meanwhile, patients with amoebic liver abscess (ALA) during the resolution phase (R-ALA) show a Th1 cytokine profile [75]. In patients with AP-ALA, a higher level of the interleukins IL-6 and IL-10, granulocyte colony-stimulating factor (GCSF), and transforming growth factor β1 (TGFβ1)] were also observed. In contrast, higher levels of IFN-γ were detected in patients with R-ALA [75].

### 3.3. Entamoeba Histolytica Migration Inhibitory Factor (EhMIF)

The *E. histolytica* migration inhibitory factor (EhMIF) is a 12-kDa protein localized to the cytoplasm of trophozoites [79]. Figure 6 depict the predicted protein structure of the migration inhibitory factor and its model confidence. It is a homolog of the human cytokine MIF [80]. This cytokine is one of the virulence factors linked to severe disease and is secreted by several medically important protozoan parasites, such as *Plasmodium, Entamoeba*, *Toxoplasma*, and *Leishmania* [81]. Structurally, EhMIF lacks signal peptides [80]. Antibodies toward EhMIF were found in the sera of children living in an area where the *E. histolytica* infection was endemic [79].

EhMIF interacts with the MIF receptor CD74 and binds to macrophages [79]. EhMIF induces IL-6 production and can also enhance TNF-α secretion. There are two ways of improving the secretion of TNF-α by inhibiting the glucocorticoid-mediated suppression of TNF-α secretion and by amplifying TNF-α production via lipopolysaccharide (LPS)-stimulated macrophages [79].

Ngobeni et al. (2017) [80] reported that mice infected with parasites overexpressing EhMIF increased chemokine expression, the neutrophil influx, and mucosal damage. In contrast, blocking the EhMIF gene decreased chemokine expression and reduced neutrophil infiltration. Furthermore, a combination of antiparasite MIF-blocking antibodies and metronidazole significantly reduced intestinal inflammation and tissue damage in mice [81]. 

In addition, Ghosh et al. (2018) [82] identified the EhJAB 1 protein as a potential binding partner and an EhMIF positive-negative regulator. The binding of EhJAB1 to EhMIF blocked the production of IL-8 by the human epithelial cells.

### 3.4. Extracellular Vesicles

The extracellular vesicle (EV) is a heterogeneous, membrane-limited structure secreted by prokaryotic and eukaryotic cells to transport lipids, proteins, and nucleic acids to the external environment [84]. They play essential roles in cellular communication, information transfer through cargo, and modulating the host’s immune system [85]. Furthermore, virulence factors and effector molecules can be transferred using EV to the host. In *E histolytica*, the secretion of EV influences encystation efficiency [84,86]. Moreover, Nievas et al. (2020) [86] suggested using EVs in vaccine studies as it has more stable conformational conditions, has the ability to circulate in bodily fluids, and uses the body's natural mechanism for transporting antigens between cells. However, thus far, no study of immune response interaction with this protein has been reported.

### 3.5. Enolase

*E. histolytica* enolase (EhENO) is one of the proteins found in the nucleus and cytoplasm of *E. histolytica* [87]. The single copy of the *E. histolytica* enolase gene (*Ehenl-1*), is located on the 1.6 Mb chromosome [88]. This protein is also predicted to have a TATA box function, as the sequence TATAAG, at −31, interacts with nuclear proteins [88]. The crystal structure EhENO manifests as an asymmetric dimer with one active site in the open conformation and the other active site in the closed conformation [89]. Both of the active sites also contain a 2-PGA molecule [89]. It catalyzes the conversion of 2-phosphoglycerate (2-PG) to phosphoenolpyruvate (PEP), which is a part of the pathway by which *E. histolytica* obtains its energy [87]. It also acts as an inhibitor of DNA methyltransferase 2 (Dnmt2) [87].

In studies by Kumarasamy et al. (2020) [57] this protein was found in the membrane fraction and was reported to be antigenic since it could be recognized by the sera of patients with amoebic liver abscesses. However, no study of human immune response interaction with this *E. histolytica* protein has been reported. There are only studies of enolase from different organisms to be found. In a study by Xue et al. (2021) [90] mice and piglets were immunized using recombinant *Mycoplasma suis* alpha-enolase (rMseno). In mice, immunizations with rMseno caused increased levels of IFN-γ and IL-4 cytokines and increased the T lymphocyte proliferation index [90]. Meanwhile, in piglets, the results showed an increased level of specific IgG1, IgG2a, CD4, and CD8 cells [90]. A study by Téllez-Martínez et al. (2019) [91] demonstrated that BALB/c mice, immunized with enolase-based vaccine and Montanide PetGel A (PGA) as an adjuvant, demonstrated a strong specific Th1 response and protective immunity against *Sporothrix schenckii*. In addition, the oral delivery of *Bacillus subtilis* spores expressing the enolase of *Clonorchis sinensis* resulted in increased specific IgM levels in sera, intestine mucus, and skin mucus in grass carps (*Ctenopharyngodon idellus)* [92]. From all the studies conducted, it can be concluded that enolase has the potential to be used as a vaccine candidate and therapeutic target, as previous studies from other organisms have demonstrated promising results. The protein structure of enolase 1, the N-terminal domain, and C-terminal Tim Barrel domain can be found in Figure 7.

### 3.6. Actin

As shown in Figure 8, actin is one of the most conserved, abundant, and ubiquitous proteins in all eukaryotes [96]. *E. histolytica* has a single actin protein, the structure of which differs significantly from human homologs. *E. histolytica* actin was reported to be antigenic and has been predicted as a suitable vaccine candidate [57,97]. It is localized in the cytoplasm and membrane of *E. histolytica* [57]. Structurally, *E. histolytica* actin contains glycine residue (Gly2) at position 2 [96]. This Gly2 is not found in most other eukaryotic actins. Still, it is crucial for cell morphology and division, host invasion in an in vitro model of human amoebic infection, and parasite-host cell adhesion [96]. Actin is an essential protein in many cellular processes, including directing the process of phagocytosis [98].

Several drug treatments have indirectly targeted actin, such as *Adenophyllum aurantium* extract, linearolactone, and kaempferol, which affects the structuration of the actin cytoskeleton [99,100]. However, until now, no immune response study has been reported regarding its interaction with *E. histolytica* actin.

A vaccine study from Li et al. (2011) reported using a DNA vaccine encoding *Toxoplasma gondii* actin against BALB/c mice [101]. Their study reported a high titer of specific antibodies and increased CD4 and CD8 T percentages, which showed that the actin could generate specific humoral and cellular immune responses [101]. Thus, we postulate that the actin from *E. histolytica* could have the same potential results as other parasites, inducing an immune response. 

### 3.7. Alcohol Dehydrogenase (ADH)

Alcohol dehydrogenase (ADH) is one of the enzymes in the cytoplasm of *E. histolytica.* It has various isoforms, but the most reported ones are alcohol dehydrogenase 2 (ADH2) and alcohol dehydrogenase 3 (Figure 9). *E. histolytica* alcohol dehydrogenase 3 (EhADH3) was reported to be antigenic in a study by Kumarasamy et al. (2020) [57], as it could be recognized by the sera of patients with amoebic liver abscesses. Some EhADH3 have been found to localize at the surface of *E. histolytica* [103]. This enzyme was also expressed at higher levels in non-pathogenic than in pathogenic amoebae [104]. The overexpression or silence of *ehadh3bb* using transfectant was found to have no or little effect on the parasite growth, size, erythrophagocytosis, motility, and hemolytic or cysteine peptidase activity [104].

Another alcohol dehydrogenase, *E. histolytica* alcohol dehydrogenase 2, called EhADH2, is a bifunctional enzyme that is essential for growth and survival [105,106,107]. It is called bifunctional because it also has aldehyde dehydrogenase (ALDH) and alcohol dehydrogenase (ADH) activities [105]. The cofactor of this enzyme, EhADH2, is Fe^2+^ [108]. Furthermore, EhADH2 is similar to other protist and bacterial bifunctional enzymes, as suggested by the phylogenetic tools [106]. This enzyme plays a role in catalyzing the conversion of acetyl CoA to acetaldehyde and the final reduction of acetaldehyde to ethanol [106,107,109]. It was also suggested by Adeoti et al. (2021) [97] as a vaccine candidate, based on bioinformatic analyses using GEPTOP CELLO NCBI. Other studies also presented it as a target for anti-amoebic inhibitors and the substrate and cofactor (NADH, Fe) binding sites could act as the inhibition targets [106]. Several studies reported that the enzymes could be inhibited by cyclopropyl (CPC) and cyclobutyl (CBC) carbinols, laboratory-tested pyrazoline derivatives, and S-nitroso-glutathione [106,110,111].

In addition, only an immune response study related to alcohol dehydrogenase was found that was not associated with *E. histolytica* but instead with *Drosophila*. Their experimental results showed that alcohol dehydrogenase played no role in the *Drosophila* immune response [112]. Thus, this protein has an equal chance of success or is not being used as a vaccine candidate. 

## 4. Docking of Membrane Protein and Vaccine Development 

The vaccine development of *E. histolytica* in studies is limited to Gal/GalNAc lectin and its components. To our knowledge, no other membrane or non-membrane proteins have been utilized as vaccine candidates. Table 1 summarizes the immune response interaction of biological membrane and non-membrane *E. histolytica* proteins and the various vaccination studies.

Docking studies elucidate how two or more molecular structures fit together to form a stable complex. It has become an important drug discovery tool and has been used for over three decades [114,115]. It is one of the most commonly used virtual screening methods, especially when the target protein's three-dimensional (3D) structure is available [114]. Aside from that, it is one of the efforts made to shorten the research timeline and reduce costs by reducing wet-laboratory experiments.

Generally, the docking of membrane proteins for vaccination involves five key stages: (1) the selection and antigenic evaluation of proteins, (2) the prediction of B-cell epitopes and T-cell epitopes, (3) the non-allergenicity and non-toxicity prediction of selected T-cell epitopes, (4) structural modeling and molecular docking, and finally, (5) an evaluation of the docking results [97,116]. The selection of proteins was conducted by prioritizing these criteria in terms of the reported antigenicity, virulence, and proteins related to the adhesion mechanisms [97]. Different databases were used, such as PATRIC3.5.16, T TMHMM v2.0, and VaxiJen V2.0. The PATRIC3.5.16 database was used to investigate the role of virulence proteins, while TMHMM v2.0 was used to predict transmembrane (TM) helixes, and VaxiJen V2.0 served to predict antigenic properties [97,116].

The second stage is the prediction of T-cell and B-cell epitopes. Identification of the MHC-I binding epitopes can be obtained using the NetMHC 4.0 server or the Propred-1 server; meanwhile, for MHC-II binding epitopes, the servers that can be used are NetMHCII 2.3 server or the Propred tool [97,116]. A server named the immune-epitope database (IEDB), which is an analysis resource, can be utilized for B-cell epitope prediction [116]. B-cell epitope prediction identifies potential antigens interacting with the B lymphocytes and initiating an immune response. The identification and selection of the B-cell epitope were conducted by referring to antigenicity, linear epitope prediction, hydrophilicity, surface accessibility, and flexibility [116].

Next, the allergenicity and toxicity of T-cell epitopes can be predicted using the AllerCatPro server and the toxpred tool, respectively. The 3D structure modeling of protein interest can be obtained from Uniprot/PDB/ProsA. The T-cell epitopes of MHC class-I peptides can be modeled using the RPBS MOBYL portal from the PEPFOLD server [116]. At this point, protein and peptide docking were then carried out. Several types of docking software can be used, including SANJEEVINI, GOLD, AUTODOCK, GemDock, Hex Protein Docking, and GRAMM. The scoring function computes scores based on the best-fitting ligand. The docking results were analyzed; the best-predicted epitopes from the study can be further tested for therapeutic potency for vaccine development.

## 5. Conclusions

To summarize, this review discusses those studies regarding the *E. histolytica* biological membrane and non-membrane protein structure, function, interaction with immune response, and recent vaccine studies. From all the membrane and non-membrane proteins, Gal/GalNAc lectin, ubiquitin, lipopeptidophosphoglycan, migration inhibitory factor, enolase, actin, ADH, and CRT interacted with the antibody. However, the extracellular vesicle was found to modulate the immune system. Furthermore, *E. histolytica* vaccine studies reported using Gal/GalNAc lectin and its components in vaccine development, such as the Gal/GalNAc lectin Lec A, LC3 fragment, and intermediate subunit Igl. Vaccine development utilizing those elements seems to have the potential to be used in combating *E. histolytica*; therefore, it should be evaluated in the clinical trial. Other membrane and non-membrane proteins can be further researched as vaccine candidates because they showed immune response interactions with the host, induced antibodies, and modulated the immune response.

## Figures and Tables

**Figure 1 membranes-12-01079-f001:**
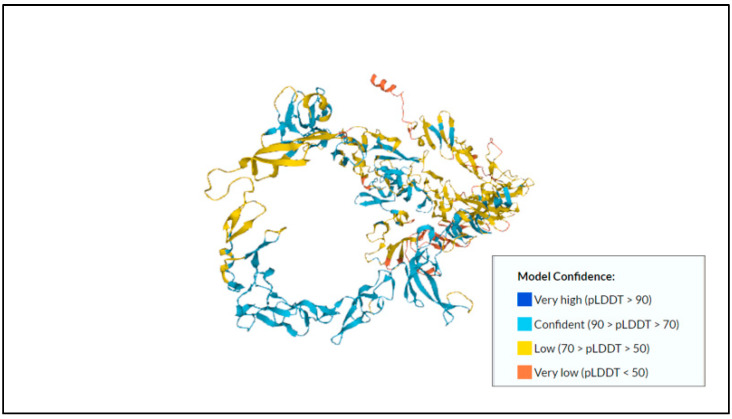
Predicted protein structure of Gal/GalNAc lectin Igl1 [40,41] (https://www.uniprot.org/uniprotkb/Q964D2/entry, accessed on 25 October 2022).

**Figure 2 membranes-12-01079-f002:**
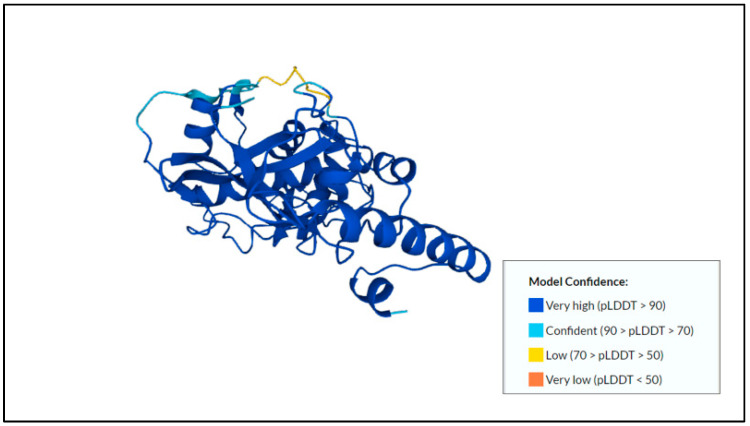
Predicted protein structure of cysteine proteinase 5 [41,49,50] (https://www.uniprot.org/uniprotkb/Q06FF8/entry, accessed on 25 October 2022).

**Figure 3 membranes-12-01079-f003:**
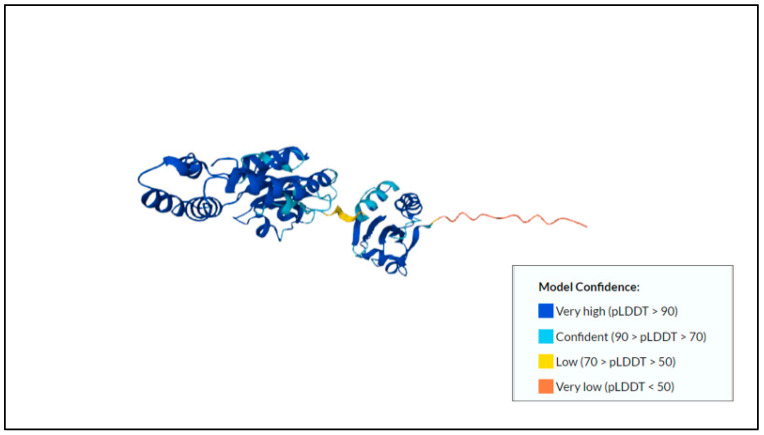
Predicted protein structure of protein disulfide isomerase [41,62,63] (https://www.uniprot.org/uniprotkb/A0A5K1UZD0/entry, accessed on 25 October 2022).

**Figure 4 membranes-12-01079-f004:**
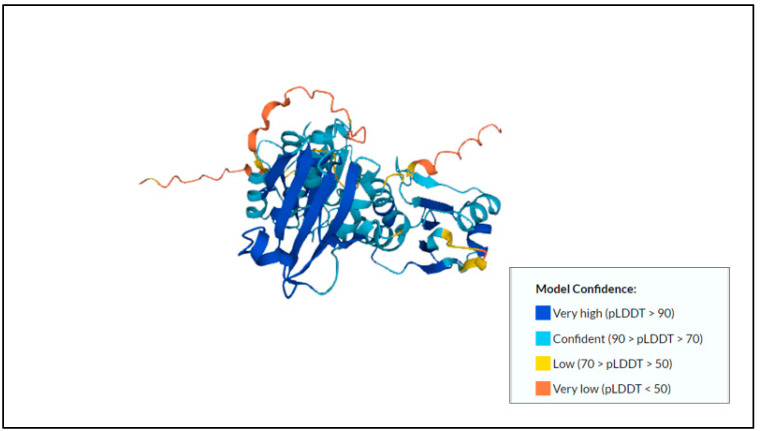
Predicted protein structure of thioredoxin (putative) [41,65,68] https://www.uniprot.org/uniprotkb/S0AYD1/entry, accessed on 25 October 2022.

**Figure 5 membranes-12-01079-f005:**
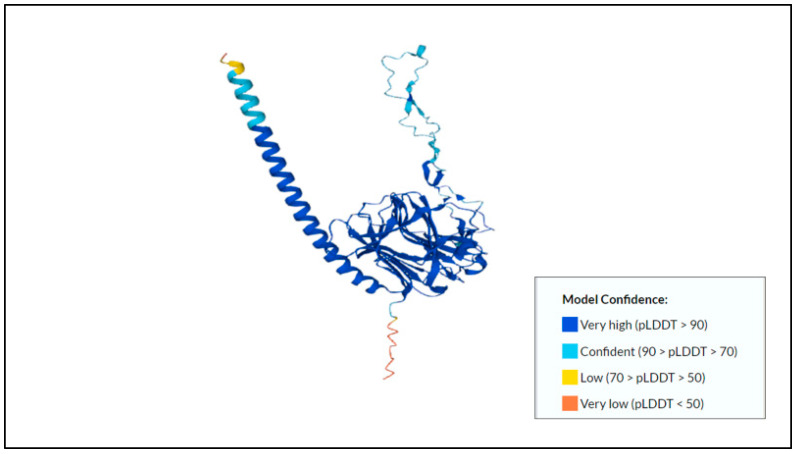
Predicted protein structure of calreticulin [41,75,78] (https://www.uniprot.org/uniprotkb/C4M296/entry, accessed on 25 October 2022).

**Figure 6 membranes-12-01079-f006:**
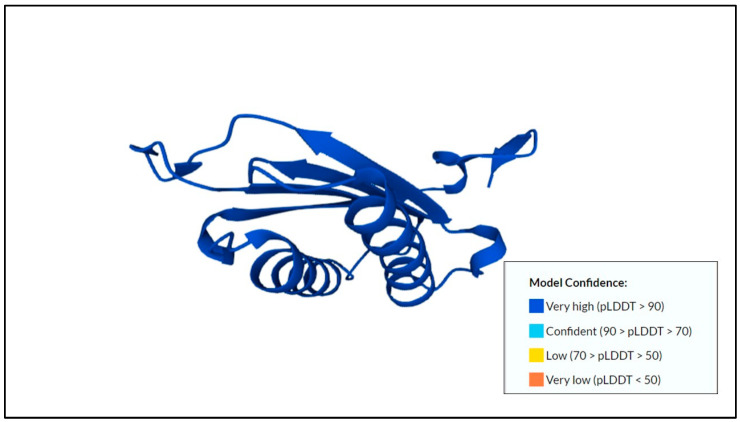
Predicted protein structure of migration inhibitory factor (EhMIF) [41,80,83] (https://www.uniprot.org/uniprotkb/A0A5K1URL6/entry, accessed on 25 October 2022).

**Figure 7 membranes-12-01079-f007:**
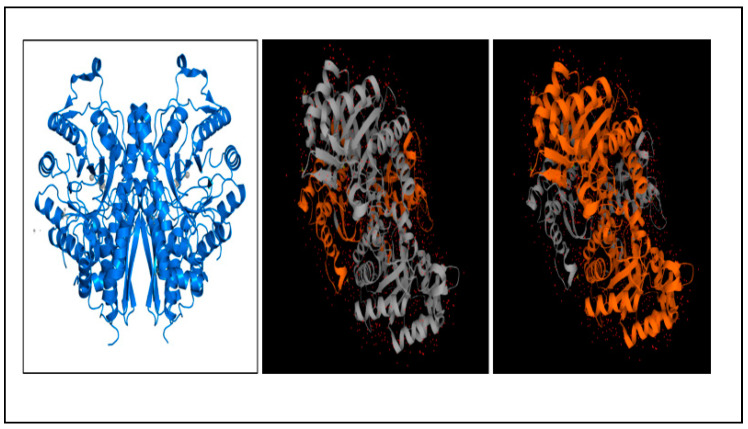
Figure showing two copies of enolase 1, N-terminal domain and C-terminal domain, respectively [89,91,92,93,94,95] (https://www.ebi.ac.uk/pdbe/entry/pdb/3qtp/analysis, accessed on 25 October 2022).

**Figure 8 membranes-12-01079-f008:**
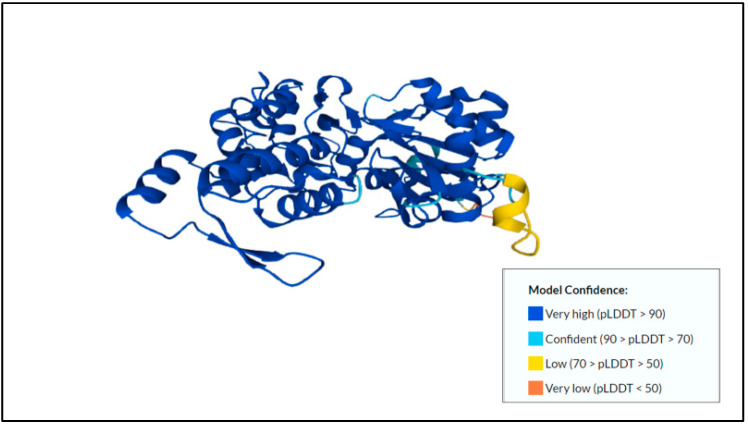
Predicted protein structure of actin [41,99,102] (https://www.uniprot.org/uniprotkb/P11426/entry, accessed on 25 October 2022).

**Figure 9 membranes-12-01079-f009:**
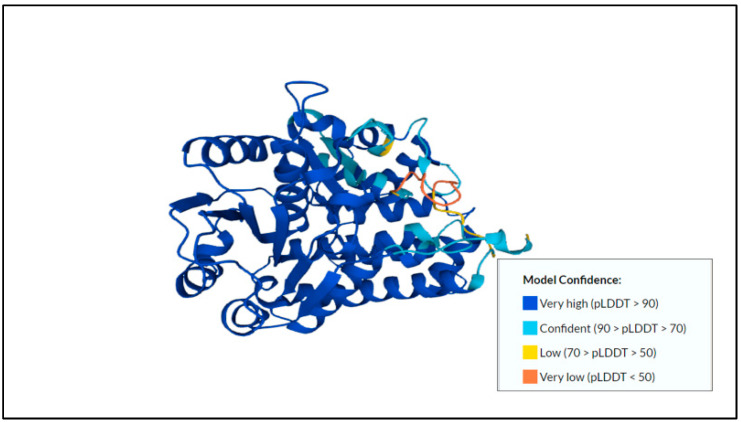
Predicted protein structure of alcohol dehydrogenase 3 [41,107,113] (https://www.uniprot.org/uniprotkb/Q24857/entry, accessed on 25 October 2022).

**Table 1 membranes-12-01079-t001:** Summary of the immune response interaction of biological membrane and non-membrane *E. histolytica* and their vaccination studies.

Protein	Types of Molecule	Antibody Interaction Study	Signal Peptide/Transmembrane Domain	Used in Vaccine Studies	References
Gal/GalNAc lectin	Membrane protein	IgA	Signal peptide	Yes	[13,15,32]
Cysteine proteinases	Membrane protein	Antigenic	Putative, transmembrane	No	[48,49]
Lipopeptidophosphoglycan (LPPG)	Macromolecule	The antibody involved was not specified	Signal peptide	No	[51,52,54,55,56]
Protein disulfide isomerase (PDI)	Membrane protein	Antigenic	Signal peptide	No	[57]
Thioredoxin	Membrane protein	Antigenic	Signal peptide	No	[57]
*E. histolytica* Ubiquitin (Ehub)	Cytosolic protein	IgG	-	No	[69]
*E. histolytica* migration inhibitory factor (EhMIF)	Cytosolic protein	The antibody involved was not specified	-	No	[79,80,81]
Enolase	Cytosolic protein	Antigenic	-	No	[57]
Actin	Cytosolic protein	Antigenic	-	No	[57]
Alcohol dehydrogenase (ADH)	Cytosolic protein	Antigenic	-	No	[57]
Calreticulin (CRT)	Cytosolic protein	Antigenic	-	No	[75,76,77]
Extracellular vesicles (EVs)	Macromolecule	Modulates the immune system	-	No	[86]

## Data Availability

Not applicable.

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
