# Peer review of "Entamoeba histolytica: Membrane and Non-Membrane Protein Structure, Function, Immune Response Interaction, and Vaccine Development"

_membranes, 2022, doi:10.3390/membranes12111079_

Round 1

Reviewer 1 Report

The manuscript by Jasni et al. entitled, “Entamoeba histolytica: Membrane Proteins Structure, Function, 2 Immune Response Interaction, and Vaccine Development” is aimed at reviewing recent advances in  E. histolytica biological membranes with the immune response for vaccination. First of all, I would like to recommend authors to design a better “Graphical Abstract” for this study to better show the whole story in a simple and informative manner. It is better to change Figs as colorful. The conclusion section is too short. Please develop it more and add future perspectives to this section as well.  I also suggest the authors devote a certain section to the “clinical application” of this technology and briefly discuss it. In addition, it is better to add some sentences about vaccination in Introduction section. 

Author Response

Dear Reviewer,

We have done the correction of the manuscript according to the comment. Please see the uploaded file.

Reviewer 2 Report

The review paper by Jasni et al., entitled “Entamoeba histolytica: Membrane Proteins Structure, Function, Immune Response Interaction, and Vaccine Development.’ Is a comprehensive review covering work to date on vaccine development for E. histolytica. The review takes a primarily protein-specific focus to cover soluble and membrane bound proteins. The review also covers  extracellular vesical components and other molecules that have been utilized in past vaccine studies.  Pictorial organization of this information would be helpful. Overall, the manuscript is well written, and data is clearly described within the manuscript. Although the authors have done an excellent job of characterizing and summarizing the literature, the presentation of information could be improved. The general readers will be interested in the summaries and findings. I would whole heartedly support the publication of the manuscript with the consideration of addressing several comments below.  

Specific Comments to Authors:

1.    Major Comment: The use of protein structures as figures for the review are interesting, but not useful for portraying addition insight for vaccine development.  The figurers would also have been better shown in color with of labeling for functional or epitopes of interest. Perhaps a table/figure of proteins involved would be more informative.

2.    Major Comment: Not everything in the table would be categorized as a protein? Table 1., could be improved by further categorizing the vaccines based on types of molecules. For example: Soluble protein, excreted protein, membrane protein or other type of molecule. 

3.    Minor Comment: Did the authors consider an overall figure for the review? That might help the reader in walking through the various sections of the manuscript and vaccine types developed to date.

Author Response

Dear Reviewer,

Thanks for your comment. We have done the correction according to the comments and suggestion. Please see the uploaded file.

Reviewer 3 Report

Dear authors,

Please see an attached file.

Author Response

Dear Reviewer,

Thanks for the comment. We have improved the manuscript writing by adding your comment.Please see the uploaded file.

Round 2

Reviewer 1 Report

The manuscript by Jasni et al. entitled, “Entamoeba histolytica: Membrane and Non-membrane Proteins Structure, Functions, Immune Response Interaction, and Vaccine Development” is aimed at reviewing recent advances in the  protein structure, function, and interaction of the biological membrane with the immune response for the development vaccines. This review became very clear for indication of  the results, procedures, and conclusions. It is better to add about docking of membran proteins for vaccination.  

Author Response

Dear Reviewer,

Please find the revised manuscript. We have addressed your comments.

Thanks

Reviewer 2 Report

The changes in the paper are excellent! The paper needs a few minor tweaks and will be ready for publication. Figure 7., is back and white and should be in color. The color schemes for the proteins should be described in the text of the figure legend or the text of the manuscript. I assume the different colors refer to the reliability of the predicted structure and folded domain? This should be clearly spelled out somewhere in the manuscript. 

Author Response

Dear Reviewer,

Thanks for the comment. We have added your suggestion into the revised version of the manuscript.

Reviewer 3 Report

Dear authors,

Thank you very much for responding to the concerns raised by the reviewer. Very nice work.

Author Response

Thank you very much